
# A data efficient machine learning model for autonomous operational avalanche forecasting

Manesh Chawla[1], Amreek Singh[2]

[1] Defence Geo-Informatics Research Establishment, Manali - 175103, India
[2] Defence Geo-Informatics Research Establishment, Chandigarh - 160037, India

*Correspondence to*: Manesh Chawla (zmfzmj123@gmail.com)

Keywords: Avalanche forecasting, hazard mitigation, random forest, machine learning, Himalaya

**Abstract:** Snow avalanches pose serious hazard to people and property in snow bound mountains. Snow mass sliding downslope can gain sufficient momentum to destroy buildings, uproot trees and kill people. Forecasting and in turn avoiding exposure to avalanches is a much practiced measure to mitigate hazard world over. However, sufficient snow stability data for accurate forecasting is generally difficult to collect. Hence forecasters infer snow stability largely through intuitive reasoning based upon their knowledge of local weather, terrain and sparsely available snowpack observations. Machine learning models may add more objectivity to this intuitive inference process. In this paper we propose a data efficient machine learning classifier using the technique of Random Forest. The model can be trained with significantly lesser training data compared to other avalanche forecasting models and it generates useful outputs to minimise and quantify uncertainty. Besides, the model generates intricate reasoning descriptions which are difficult to observe manually. Furthermore, the model data requirement can be met through automatic systems. The proposed model advances the field by being inexpensive and convenient for operational use due to its data efficiency and ability to describe its decisions besides the potential of lending autonomy to the process.

## 1. Introduction

In snow bound mountainous areas worldwide, avalanches cause significant loss of life and property. Avalanche deaths are estimated at 250 per year (Schweizer et al., 2015). Government and private agencies are funded to reduce avalanche risk for important activities and property e.g. road/rail transport, winter sports, construction, military operations etc. This effort has led to development of several techniques to reduce avalanche risk. Hazard mapping is done to estimate the long term hazard at each avalanche path in a region (Choubin et al., 2019; Rahmati et al., 2019). The map is used to implement active risk reduction measures e.g. building control structures, modification of nearby terrain or use of explosives to trigger avalanches in controlled way (Fuchs et al., 2007). Using active methods at each hazardous path is economically infeasible therefore avalanche forecasting is practised to reduce exposure to avalanches. Individuals can use information in forecast to minimise risk in short term.

Avalanche forecasting aims to identify the locations of snowpack weakness, their spatial distribution, sensitivity to triggering and the size of resulting avalanches (Statham et al., 2018). Snow stability shows high variance with respect to terrain features (Gaume et al., 2014). Also, observing snowpack stability of a large area at high spatio-temporal resolution is difficult. Therefore stability at most of the avalanche slopes is deduced using secondary observable data e.g. meteorological and snowpack parameters from a similar representative site, terrain parameters of the slope, expected changes to snowpack by imminent weather etc. Deduction process for snow stability from secondary data is yet to be satisfactorily formularized mathematically. Therefore, forecasters need to rely on their intuition of local terrain and snowpack patterns to estimate stability and collect more information to minimise uncertainty (LaChapelle, 1980; Schweizer et al., 2008; McClung and Schaerer, 2006). Numerical (physical-based), statistical, heuristic and machine learning models are important tools for adding objectivity to this process.


Physical-based numerical models such as CROCUS (Brun et al., 1992; Vionnet et al., 2012)and SNOWPACK(Bartelt and Lehning, 2002)simulate the snowpack and weather processes that contribute significantly to avalanche hazard. These models give accurate snow profile simulations at microscale level (<1km$^2$) for avalanche paths where meteorological data

is available. Since, meteorological sensors cannot be placed at all hazardous paths therefore interpolated meteorological data from tools like SAFRAN and meteoIO are used as input for numerical model to simulate the snowpack (Durand et el., 1999; Bavay and Egger, 2014; Morin et al., 2020). Output from CROCUS or SNOWPACK shows forecasters the changing state of snowpack due to numerically modelled processes, viz., weak layer formation due to temperature gradients, surface or deep wetting, compaction and refreezing etc. However, its accuracy can be seriously affected by errors in interpolated

meteorological data.

Avalanche forecasting models based upon statistical techniques such as discriminant analysis, regression trees, hidden Markov models (HMM) and k-nearest neighbours (k-NN) (Obled and Good, 1980; Buser, 1983; Davis et al., 1999, Gassner et al., 2001; Schirmer et al., 2009) use input from a specific location to represent snow and weather conditions of a larger region (generally mesoscale ~ 100 km$^2$). These models link weather and snowpack variables to hazard using

avalanche occurrences from historical data. Information from multiple sources (possibly redundant) e.g. wind loading indexes, local terrain features (slope, elevation etc.), location specific snowfall patterns, numerical snowpack simulations and numerical weather model output can be included in these models (Singh et al., 2005; Schirmer et al., 2009; Bellaire et al., 2017). This makes them more robust to errors in individual parameters compared to numerical models. However, they don't directly model the inductive reasoning process used by avalanche forecasters. Therefore such models use the training

data inefficiently and require larger training samples to achieve a specified performance. Models by Buser (1983), Gassner et al. (2001), and Singh et al. (2005) etc. require at least seven years' training data. Also, some features used in above-referred implementations require manual effort to record. Interpreting model output may also be difficult for a forecaster in most cases. In this regard, k-NN models indeed havepossibility of generating descriptive output in terms of list of events and its incorporation into decision making process similar to conventional inductive avalanche forecasting processes

(LaChapelle, 1980).

Heuristics based expert systems have been attempted to model complex patterns which are missed by nearest neighbours (Schweizer and Föhn, 1996). This approach is capable of using expert knowledge by modelling known forecasting rules. But considerable human effort and expertise is required to build such models.

Machine learning has been quite successfully used for tasks where procedures cannot be precisely formulated but humans
perform well e.g. in handwriting and speech recognition (Liang and Hu, 2015). Machine learning models are therefore used for several tasks supporting avalanche risk mitigation. Classification and regression trees (Davis et al., 1999; Rosenthal et al., 2001; Hendrikx et al., 2005; Hendrikx et al., 2014) have been explored for assistance of avalanche forecasters as these techniques provide easily comprehensible interpretations of complex interactions. Support vector machine (SVM) was implemented as a data exploration tool and a predictive engine for spatio-temporal forecasting of snow avalanches

(Pozdnoukhov et al., 2008; Pozdnoukhov et al., 2011).Purves et al. (2003) and Singh et al. (2015) applied nature-inspired meta-heuristics to optimize weights assigned to input variables of $k$-NN models. Singh and Ganju (2008) exploited artificial neural networks (ANN) for post processing of a $k$-NN model output for improved classification ability. Rubin et al. (2012) explored avalanche detection from seismic signals using machine learning models. Dekanová et al. (2018) proposed use of fast ANN to process data from automatic weather station for assistance in determining the avalanche

danger levels. Choubin et al. (2019) and Rahmati et al. (2019) applied and validated machine learning techniques for avalanche hazard mapping.

In this paper we propose a machine learning model based on random forest (RF) technique (Opitz and Maclin, 1999; Breiman, 2001)for avalanche or no-avalanche classification using snow and weather parameters. In RF, an ensemble of





decision trees (constructed by automatic rule inference) generates the prediction and quantifies the uncertainty. RF
ensemble can learn complex decision boundaries and is resistant to over-fitting (Breiman, 2001) leading to better
generalization of the model and in turn ensuring high quality results. The model is inexpensive and convenient to use for
operational applications due to its data efficiency and interpretable data mining outputs. Besides, the input data can be
observed using automatic devices. These properties can be leveraged to set-up an autonomous avalanche forecasting
system with slight human intervention.

The paper is structured as follows. Section 2 introduces the RF technique. Section 3 describes the feature set and study
area. Model training and generalization performance has been discussed in Section 4. Section 5 explains how descriptive
output of model can be effectively used by forecasters to validate their intuitions. In Section 6, the salient features of the
model have been compared with other models in use. Section 7 highlights the data efficiency aspect and potential of using
the model for setting up autonomous decision making frameworks. The conclusions of the study and future research
potential in the field are listed in Section 8.

## 2. Random Forest Technique

RF is an ensemble learning method (Opitz and Maclin, 1999) where an ensemble of decision trees generates the prediction.
A decision tree describes a flowchart like process for classifying the query. Each tree node represents a step in the process.
An internal (non-terminal) node defines a test condition. Outgoing arrows from an internal node represent future steps
depending on the outcome of test at the node. The arrow connected nodes are called the children nodes of the internal node.
Terminal step is represented by leaf nodes i.e. nodes without any children. To classify a query, the test is applied to the
query at root node and depending on the result a child node is selected. If child node is an internal node, the same process
is repeated to move to subsequent child node. This is repeated till a leaf node is reached. This leaf node defines the
classification for the query. Algorithms for training (also called 'learning' or 'construction') of a decision tree proceed by
splitting the given training dataset based on a feature value such that the resulting split datasets are more homogeneous in
their target variable. This splitting process continues recursively on datasets till termination criteria is reached which
specifies that the each split dataset is sufficiently homogenous. Recursive splitting process naturally defines the decision
tree. Each node corresponds to a dataset and the split mentioned in node corresponds to the split decided by the training
process. The highest frequency class label of the final homogenous split datasets is taken as the label represented by the
corresponding leaf node.

In this work, we used C4.5 algorithm (Bressert, 2012; Quinlan, 1993) based RF classifier implemented in scikit-
learn[1] (Pedregosa et al., 2011) for decision tree constructions. C4.5 splits on an attribute with highest normalised
information gain, a measure based on the concept of entropy from information theory. The information content $I(E)$ (also
called the *surprisal*) of an event $E$ is defined as $I(E) = -\log_2 p(E)$ where $p(E)$ is probability of occurrence of $E$. The choice
of base varies between different applications: base 2 gives the unit of *bits* (or "shannons"), while base $e$ gives the natural
units *nat*. Entropy measures the expected (i.e., average) amount of information conveyed by identifying the outcome of a
random trial. In a dataset $T$, entropy $H(T)$ is defined as (*Witten, 2011)*:

$$H(T) = I_E(p_1, p_2, ..., p_J) = -\sum_{i=1}^{J} p_i \log_2 p_i \tag{1}$$

where $p_1, p_2, ...$ represent the percentage of each class present in $T$. When $H(T) = 0$, the set $T$ is perfectly classified (i.e. all
data points in $T$ are of same class). Information gain $IG(T, a)$ is the measure of the difference in entropy from before to
after the set $T$ is split on an attribute $a$.

---

[1]An open source machine learning software library for the Python programming language (https://scikit-learn.org/stable/)





$$IG(T, a) = H(T) - H(T|a)$$
$$= H(T) - \sum_{S \in T} p(S) H(S) \tag{2}$$

Where,

$S$: the subsets created from splitting of set $T$ by $a$ such that $T = \bigcup_{S \in T} S$,

$p(S)$: the proportion of the number of elements in subset $S$ to the total number of elements in set $T$, and

$H(S)$: Entropy of subset $S$

In order to inhibit the tendency of the classifier to be biased towards the majority class while dealing with the problems characterized with imbalanced data, Chen et al. (2004) proposed to place a heavier penalty on misclassifying the minority class by assigning a weight to each class, with the minority class given larger weight (i.e., higher misclassification cost). The scikit-learn RFclassifier has the option ("balanced" mode) to introduce

such weighted cost for the classes to take care of imbalanced data. This modifies the computation of $p_i$ and $p(S)$ in Eqn. (1)&(2) respectively as follows:

$$p_i = \frac{w_i}{\sum_k^J w_k} \tag{3}$$

$$p(S) = \frac{Sum\ of\ weights\ of\ elements\ in\ S}{\sum_k^J w_k} \tag{4}$$

Where $w_i$ is the weight of $i^{th}$ class. The "balanced" mode automatically adjusts weights inversely proportional to class frequencies in the input data.

Information gain (Eqn. (2)) is normalized to penalise excessive splitting at a node. The normalisation factor is the intrinsic

value defined as

$$IV(T, a) = - \sum_{S \in T} p(S) \log_2 p(S) \tag{5}$$

And therefore

$$Normalised\ Gain\ (T, a) = \frac{IG(T, a)}{IV(T, a)} \tag{6}$$

Individual decision trees are sensitive to small changes in data and unable to learn complex decision boundaries without overfitting (Hastie et al., 2009). In RF, each tree of ensemble is constructed on an independent random subspace derived

from the training dataset using a process called bagging. This ensures that individual trees are uncorrelated (Breiman, 1996). A subspace is formed by drawing a random sample dataset with replacement from training set and then selecting a random subset of features from the drawn sample. To build the ensemble a user specified number of decision trees are constructed and stored in memory. For a given query, output of the ensemble is the mean output of individual trees in terms of probability values calculated as the proportion of the target class present in terminal leaf node split dataset.


**3. Study Area and Data Characteristics**

The proposed model has been trained and tested using snow-meteorological and avalanche occurrence observations from Bandipore-Gurez (BG) sector at the tip of Great Himalayan Range in the north-western part of Indian Himalaya (Figure 1). In this area, a major highway runs along the Kishenganga river in Gurez valley and Tulel valley and connects to Bandipore

town in Kashmir valley through Razdan pass (3300m above m.s.l.). In Gurez valley (area on west of Wampore town), 40 major avalanche paths affect the highway stretch of about 25 kms from Jatkushu village to Wampore village. Besides, about 15 avalanche paths affect the lateral tracks. In Tulel valley (area on east of Wampore town), over 100 major and


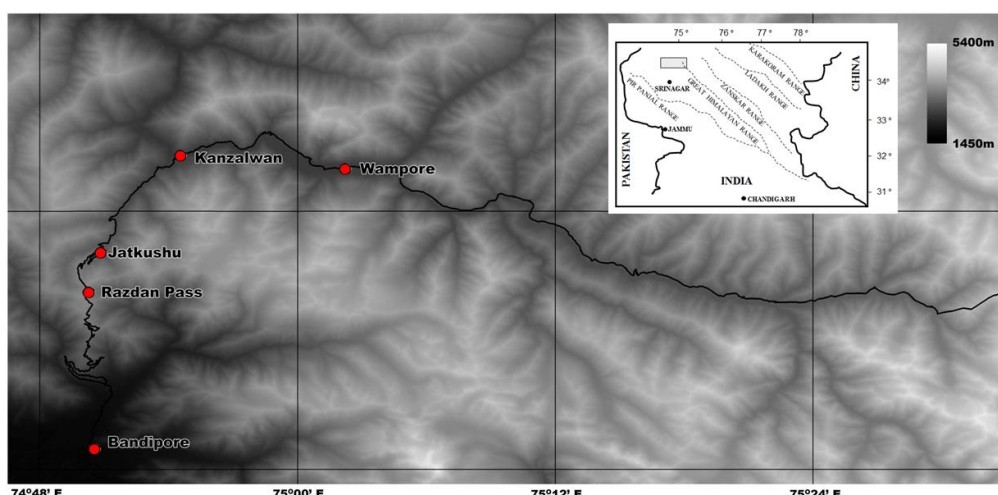

**Figure 1: A perspective view of Bandipur-Gurez sector**

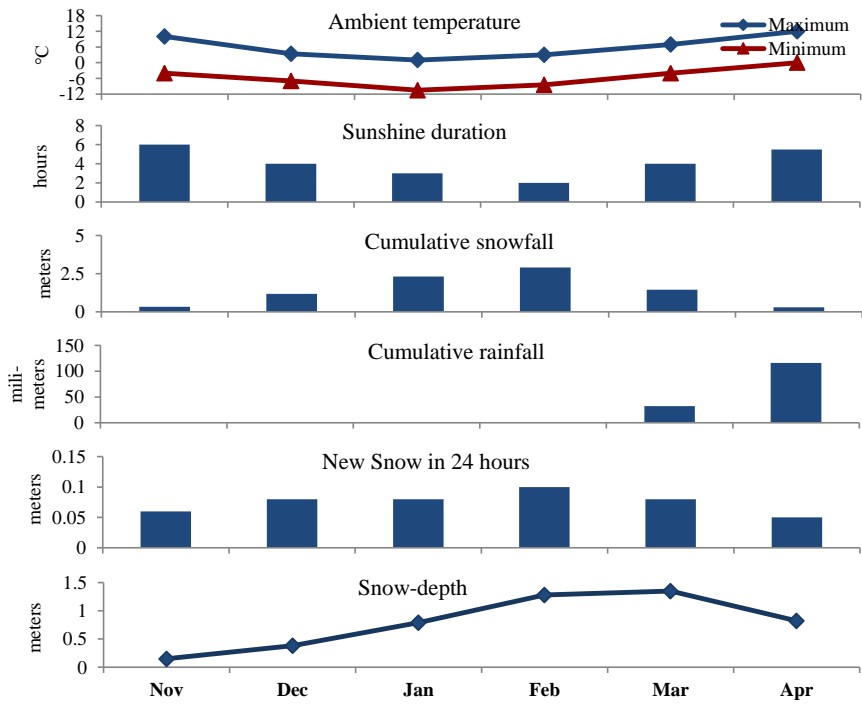

**Figure 2: Summary of snow-meteorological observations from BG sector (The values shown are medians of respective variables in different months over a period from Nov-1993 to April-2017)**

minor avalanche paths affect the highway and lateral tracks. The start-zone elevation of these avalanche paths ranges from about 2350m to 4800m above m.s.l. A snow-meteorological observatory is located near Kanzalwan town (2440m above

m.s.l.). A summary of winter-season observations (for the months from November of a year to April of next year) from the observatory is presented in Figure 2. The values shown are the medians of monthly observations of respective parameters over a period from November-1993 to April-2017. According to the algorithm proposed by Mock and Birkeland (2000),the



**Table 1: Intrinsic snow-meteorological features used by model. All features are recorded at 08:30 IST[*].**

| Parameter name | Unit | Description |
|---|---|---|
| MAX_TEMP | °C | Maximum Temperature of past 24 hours |
| MIN_TEMP | °C | Minimum Temperature of past 24 hours |
| SNOW_TEMP | °C | Snow surface temperature |
| SNOW_HEIGHT | m | Height of snow surface above ground level |
| NEW_SNOW | m | New snowfall in past 24 hours |
| WIND | m/s$^{-1}$ | Average wind speed in past 24 hours |
| AVAL | - | Number of avalanches triggered in the area in past 24 hours |

[*]IST: Indian Standard Time (UTC + 05:30)

**Table 2: Features derived from intrinsic features as described in Table 1.**

| Parameter name | Unit | Description |
|---|---|---|
| SNOW_TEMP_DIFF | °C | Snow surface temperature difference from past day |
| NEW_SNOW2 | m | Cumulative new snowfall in past 2 days |
| NEW_SNOW4 | m | Cumulative new snowfall in past 4 days |
| NEW_SNOW10 | m | Cumulative new snowfall in past 10 days |
| WIND2 | m/s$^{-1}$ | Average wind speed of past 2 days |
| WIND4 | m/s$^{-1}$ | Average wind speed of past 4 days |
| WIND10 | m/s$^{-1}$ | Average wind speed of past 10 days |
| AVAL2 | - | Number of avalanches triggered in the area in past 2 days |
| AVAL4 | - | Number of avalanches triggered in the area in past 4 days |

snow climate of BG sector may be classified as continental as the area witnesses low air temperature and high snowpack temperature gradient in early December.

The input data features used for model development are summarized in Table 1 and Table 2. While features in Table 1 are direct snow-meteorological observations, features in Table 2 are derived from those in Table 1 and recent avalanche activity in the area to represent the conditions of past few days. The technology for automatic observation of features included in Table 1 has already been in existence for many decades now. Recently infrasonic and seismic sensors as well as Doppler radar based systems have been demonstrated to automatically detect and locate avalanche occurrences also

(Rubin et al., 2012; Thuring et al., 2015;Persson et al., 2018). Thus all the necessary input data for the proposed model may be recorded using automatic devices.

## 4. Model Setup and Performance Analysis

### 4.1 Data pre-processing

In order to demonstrate that the proposed model is data efficient, data of just three winter seasons (from December 2010 to March 2013) only was used to construct the ensemble. Let us call it 'training data'. Thereafter, model was tested on 'test data' from four subsequent winter seasons (December 2013 to March 2017). The data comprised of values of features summarized in Table 1-2. Each data row also carried a class label$\in \{0,1\}$(0: No-avalanche day and 1: Avalanche day). The initial dataset contained more no-avalanche day cases (about 75%) than avalanche day cases (about 25%). A classifier

trained on this skewed dataset will be biased to forecast more no-avalanche days. A popular approach to deal with such issues is to use cost corrected classifiers with higher cost assigned to minority examples. Another approach is to discard majority class data randomly or synthetically generate more minority class data to make class sizes comparable. However, this approach can lead to over fitted classifier. We rather opted to apply domain specific knowledge to deal with this issue





**Table 3: Summary statistics of BG sector avalanche dataset (Dec 2010 – Mar 2013). Avalanches are unlikely when SNOW_HEIGHT < 0.50m.**

| | |
|---|---|
| Number of days in a winter season* | 121 |
| Mean number of avalanche days per season | 32 |
| Mean number of days per season with SNOW_HEIGHT > 0.50m | 87 |
| Mean number of avalanche days per season with SNOW_HEIGHT > 0.50m | 31 |

\* From 01-December of a year to 31-March of following year

**Table 4: Model performance measures** (Wilks, 1995). $i, j \in \{0, 1\}$ **where 0: No-avalanche day and 1: Avalanche day.**

| Measure name | Description | Expression in terms of confusion matrix |
|---|---|---|
| False Alarm Rate (FAR) | Conditional probability of returning an avalanche day given underlying day is no avalanche day | $\dfrac{a_{10}}{a_{10} + a_{00}}$ |
| Probability of Detection (POD) | Conditional probability of forecasting an avalanche day given underlying day is avalanche day | $\dfrac{a_{11}}{a_{11} + a_{01}}$ |
| Precision | Fraction of predicted days which are avalanche days | $\dfrac{a_{11}}{a_{11} + a_{10}}$ |
| Heidke Skill Score (HSS) | Measures the forecast performance of classifier over of a defined random forecast | $\dfrac{2(a_{11}a_{00} - a_{10}a_{01})}{(a_{11} + a_{01})(a_{01} + a_{00}) + (a_{11} + a_{10})(a_{10} + a_{00})}$ |
| Hansen Kuipers Skill Score or True Skill Score (TSS) | Measures the forecast performance of classifier over of a defined random forecast | $\dfrac{a_{11}a_{00} - a_{10}a_{01}}{(a_{00} + a_{10})(a_{01} + a_{11})}$ |

of skewness in data. We removed all such cases from training dataset for which avalanches are unlikely due to lack of

sufficient snowcover (Canadian Avalanche Association; 2016). Particularly, we discarded cases with SNOW_HEIGHT< 0.50m. This filtering step removes poor examples which can decrease model performance. See Table 3 for justification of threshold choice for filtering and summary statistics of the dataset. Besides, when training decision trees of ensemble, the classes are weighted inversely to their proportion in filtered dataset to place a heavier penalty on misclassifying the minority class in order to counter imbalance in data (section 2). One obvious consequence of above-mentioned filtering

on data is that for all the queries with SNOW_HEIGHT< 0.50m,classifier would not be applied.

**4.2 Performance measures**

Let us define a confusion matrix of a classifier $C$ as on a labelled dataset $D_s$ as:

$$a_{ij} = |S_{ij}| \tag{7}$$

Where $S_{ij}$ is defined as:

$$S_{ij} = \{x \in D_s : C(x) = i \text{ and } label(x) = j\} \tag{8}$$

This matrix is used for performance analysis of classifiers. Here we derive the performance measures from confusion matrix $a_{ij}$ with $i, j \in \{0, 1\}$ (0: No-avalanche day and 1: Avalanche day) as summarized in Table 4 to describe model performance.



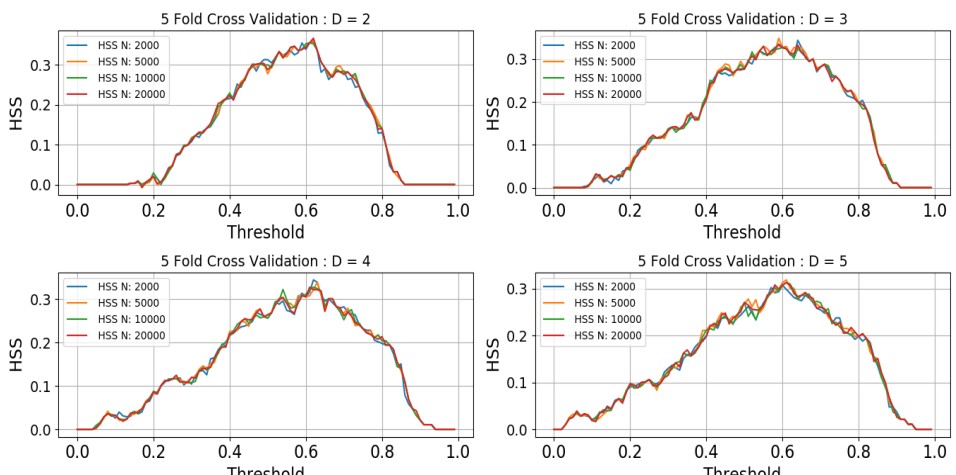

**Figure 3: HSS–Threshold probability curves (5-fold cross-validation on training data) for different values of *N* and *D*.**

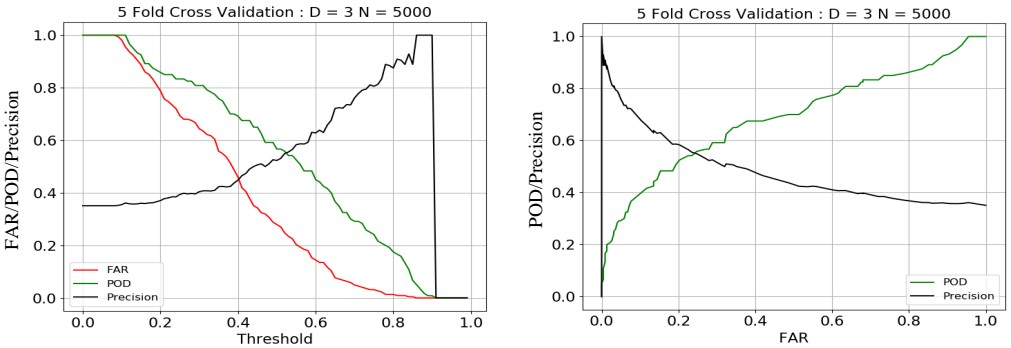

**Figure 4: FAR, POD and Precision curves against threshold probability (5-fold cross-validation on training data) for *D* = 3 and *N*=5000.**

### 4.3 Model training and hyper-parameters tuning

A classification model is trained and cross-validated with hyper-parameters tuned for optimized performance. The hyper-parameters tuning is an expert task and may involve a few trial experiments. There are following two hyper-parameters in RF technique:

*D* : Maximum depth allowed (number of features considered) for each tree in the ensemble

     *N*: Number of trees used in ensemble

RF model outputs the estimated probability of an avalanche against a query expressed in terms of input variables defined in Table 1-2. To convert the probability output to a binary classification, a threshold value for probability is fixed. If probability prediction for a query is greater than the threshold, it is classified as an avalanche day (class label = 1)

otherwise a no-avalanche day (class label = 0). The choice of threshold sets a trade-off between the risks of missing an avalanche day against the risks of false alarm on a no-avalanche day. Low threshold values give high false alarm rates but fewer avalanche days are misclassified as no-avalanche days. High threshold choice gives few false alarms (improves precision) but misses more avalanche days. Forecaster can select a threshold optimal for his risk management strategy without retraining the model. Thus for any given set of values of hyper-parameters, the performance of RF model in terms



of binary outputs will also vary with choice of threshold probability. Based on this premise, model training (i.e. ensemble construction) experiments were conducted with 5-fold cross-validation using training data (refer section 4.1) with all possible combinations out of following values of hyper-parameters for search of optimal set of values:

$D$: 2, 3 4, and 5

$N$: 2000, 5000, 10000, and 20000

Trees tend to over-fit for large values of $D$. But for $N$, large values are preferred because the output of ensemble is the mean of individual tree outputs and as $N$ increases the variance of mean output decreases and decision boundaries become smoother (Breiman, 2001). This explains the above-mentioned choices for search of optimal values of $D$ and $N$.

The experimental results in terms of HSS (refer Table 4) against varying threshold probability values were analysed. These results have been obtained with thresholds uniformly spaced between 0 and 1 with gap of 0.01. It was observed that there is

no significant change in the performance as measured by HSS by varying the value of $N$ from 2000 to 20000 for any particular values of $D$ (refer Figure 3). Also increasing values of $D$ beyond 3 decreased the HSS on most threshold

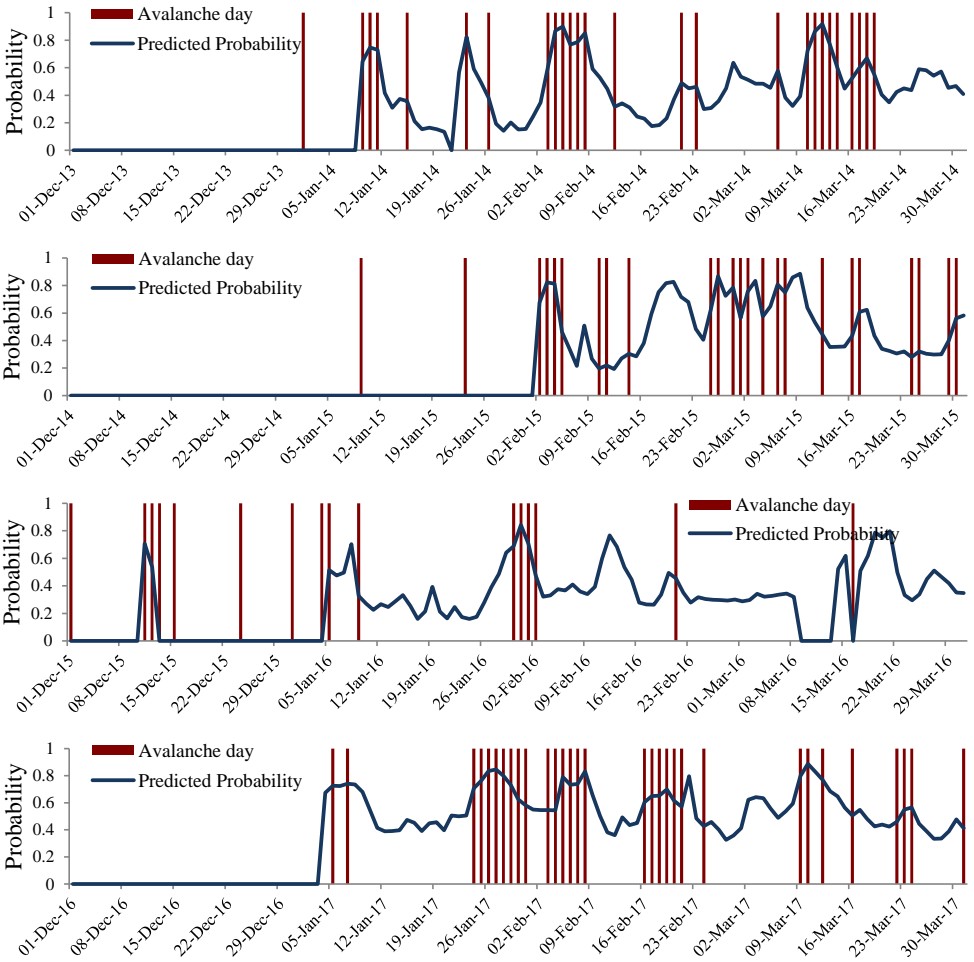

**Figure 5: Model prediction results against observed avalanche activity (by default, for all the cases with SNOW_HEIGHT < 0.50m the prediction probability value is set as 0)**

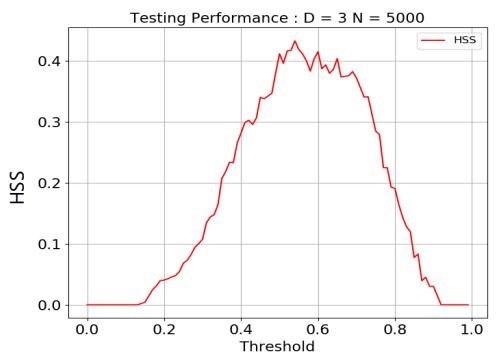

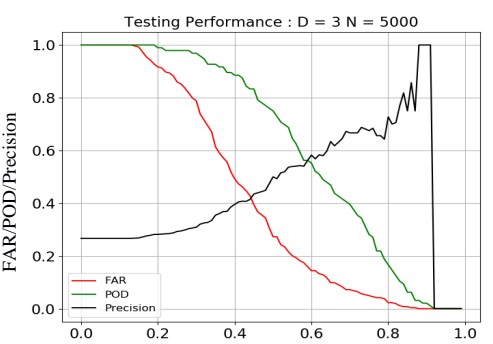

**Figure 6:** HSS–Threshold probability curve (testing phase) with *D*=3 and *N*=5000

**Figure 7:** Precision, POD and FAR for various thresholds (testing phase) with *D*=3 and *N*=5000

**Table 5: Comparison of cross-validation and generalization performance at various FAR levels**

| FAR | Cross-validation | | | Generalization | | |
|---|---|---|---|---|---|---|
| | POD | Precision | HSS | POD | Precision | HSS |
| 0.2 | 0.53 | 0.58 | 0.33 | 0.65 | 0.54 | 0.42 |
| 0.3 | 0.59 | 0.51 | 0.28 | 0.76 | 0.47 | 0.38 |
| 0.4 | 0.68 | 0.47 | 0.25 | 0.83 | 0.43 | 0.34 |
| 0.5 | 0.70 | 0.42 | 0.19 | 0.88 | 0.40 | 0.28 |
| 0.6 | 0.78 | 0.41 | 0.14 | 0.91 | 0.36 | 0.21 |
| 0.7 | 0.83 | 0.39 | 0.09 | 0.93 | 0.33 | 0.15 |

probability choices. Hence, we chose $D = 3$ and $N = 5000$ as the values of hyper-parameters of the ensemble for the study area. Training with these values requires low computational resources and still ensures model convergence.

Figure4 shows 5-fold cross-validation results in terms of FAR, POD and Precision scores for the above-selected hyper-
parameters values. High classification threshold probability means only days when the model is highly confident are classified as positive (avalanche days). This is well reflected in Figure 4 where increasing threshold improves precision and lowers false alarms. However, beyond threshold of 0.92 precision is zero, because 0.92 is the highest predicted probability on the training data. As the value (= 0.92)of highest predicted probability is significantly lower than the perfect value of 1, this is a shortcoming of ensemble forecast in theoretical sense.

Rate of change of POD with respect to FAR is high for lower values of FAR (say less than 0.2). In contrast, allowing higher FAR beyond a certain value (say beyond 0.6) gives smaller gains in POD. Avalanches may occur due to complex situations not represented by training data. Therefore the model is not confident when avalanches occur due to such reasons. Simpler situations e.g. high NEW_SNOW and high SNOW_HEIGHT can be easily detected by model at lower FAR. Predicting complex situations demand higher FAR. The details presented in Figure 4 along with corresponding HSS–
Threshold probability curve from Figure 3can be used to choose a reasonable FAR-HSS trade-off and fix the threshold probability value for operational use of the model.

**4.4 Model generalization**

After training, the ability of a machine learning model to generalize (response to new data) well is central to its success. So
after having tuned the hyper-parameters through cross-validation experiments for optimized performance as explained in previous section, we constructed the ensemble using entire training data and then tested its generalization ability using test


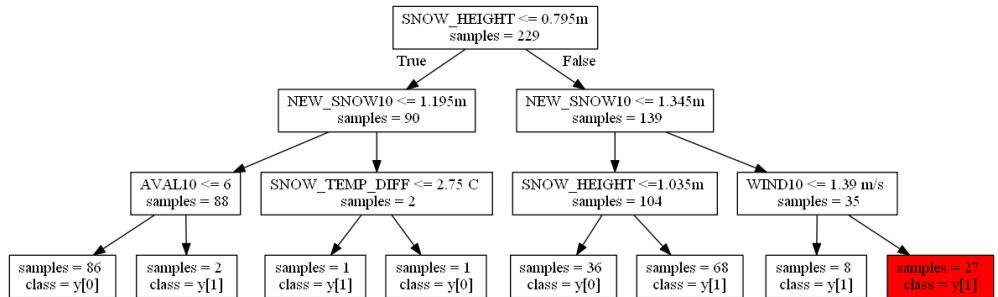

**Figure 8: Decision Tree demonstrating contributing factors on 01–Feb–2017 at BG sector, selected from ensemble for visualisation due to its high probability output.**

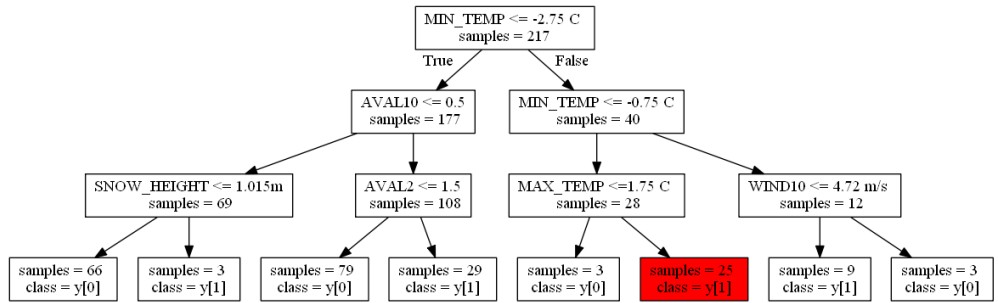

**Figure 9: Decision Tree demonstrating contributing factors on 28–Mar–2017 at BG sector, selected for visualisation from ensemble due to its high probability output. The tree indicates that melting maybe a major reason for threat. Numerical thresholds obtained can be helpful for further data mining.**

data (refer Section 4.1). The prediction probability values corresponding to each day of the test period against expected probability values (1: avalanche day, 0: no-avalanche day) are presented in Figure 5.The performance has also been evaluated using scores described in Table 4 with thresholds uniformly spaced between 0 and 1 with gap of 0.01 (Figure 6-

245    7).

We find that performance scores obtained against test data(Figure 6-7) are even better than what we obtained during cross-validation experiments (Figure 3-4). A comparison of performance scores obtained from cross-validation and those obtained from generalization exercise against a range of FAR values is presented in Table 5 for easy comprehension. It testifies that the constructed ensemble has good generalisation ability.


### 5. Model Output Interpretability

Machine learning based methods are often characterized as 'black-boxes'. RF model addresses this aspect by generating descriptive outputs. Such descriptive outputs help forecasters to understand the reasoning behind the decision taken. This information is also useful to find particular unstable slopes as well as to estimate the type and magnitude of avalanches. In

a decision tree, a path from root to a leaf node can be interpreted as a sequence of conditions defining the forecasting rules. Out of the ensemble of trees, the subset of trees predicting highest avalanche probabilities represent logic applied to current situation and the strength of its predictive value. In our experiments, the trees have shown non-trivial reasoning which may be difficult to discover otherwise. As an example, for 01-Feb-2017 predicted ensemble mean probability was 0.54.This




**Table 6: Verification of decision tree output by comparing statistics of data filtered using tree output and unfiltered/control datasets ($n$: sample size).**

| Statistic | Unfiltered Dataset | Dataset filtered by temperature bounds | Dataset filtered by SNOW_HEIGHT> 1.0 m |
|---|---|---|---|
| Proportion of Avalanche days | 0.21 ($n = 849$) | 0.43 ($n = 95$) | 0.4 ($n = 330$) |
| Mean SNOW_HEIGHT | 0.81m ($n = 849$) | 1.01m ($n = 95$) | 1.45m ($n = 330$) |
| Proportion of Avalanche days when $0 \leq$ NEW_SNOW $\leq 0.2$ m | 0.27 ($n = 194$) | 0.61 ($n = 29$) | 0.45 ($n = 95$) |
| Proportion of Avalanche days when $0.2$ m $\leq$ NEW_SNOW $\leq 0.4$ m | 0.41 ($n = 70$) | 0.68 ($n = 16$) | 0.5 ($n = 44$) |
| Proportion of Avalanche days when NEW_SNOW> 0.4 m | 0.51 ($n = 36$) | 0.71 ($n = 6$) | 0.58 ($n = 24$) |

value indicates high snowpack instability as avalanches have triggered in at least about 50% of cases when predicted mean probability was $\geq 0.5$ (refer Figure 7 where for threshold probability $\geq 0.50$, the precision is $\geq 0.49$). Corresponding to this case, ten decision trees with predicted probabilities $\geq 0.9$ were analysed. Most trees show that snowfall in past 10 days and high wind speed caused hazard. Tree in Figure 8 demonstrates this reasoning pattern. Following the reasoning path from root to terminal leaf node (shown in red colour) we get the following heuristic satisfied:

*IF* (SNOW_HEIGHT>0.795m) *AND* (NEW_SNOW10>1.345m) *AND* (WIND10 >1.39m/s)

*THEN* (avalanche probability>0.90)

Such reasoning is known to experienced forecasters. In this case model gives numerical estimates for intuition. Trees also suggest patterns which are difficult for forecasters to observe manually. Figure 9 demonstrates such a pattern. This was visualised for case of 28-Mar-2017.Suggested rule satisfied for the day is:

*IF*(-2.75 ºC< MIN_TEMP $\leq$ -0.75ºC) *AND* (MAX_TEMP $\geq$ 1.75ºC)

*THEN* (avalanche probability >0.90)

The above rule about temperature bounds suggests that snow melt maybe causing hazard and wet avalanche is likely. Such a simple yet effective rule in terms of temperature only is difficult to find for a forecaster. Notwithstanding the observation, other features correlated to the temperature bounds may actually be causing hazard. To rule such a possibility out, we made a simple univariate analysis, where variables with significantly different distributions in temperature filtered and original datasets were analysed. To analyse effect of snow height, we applied another filtering to get data where snow height is greater than the mean snow height of temperature filtered data. Statistics from these three datasets are compared in Table 6. The data mining results in Table 6 show that when the temperature bound rule is satisfied snowfall leads to higher triggering probability. This is due to combination of factors: formation of melt-freeze crusts and higher density of fresh snow at higher temperatures (Statham et al., 2014; Meløysund et al., 2007). The fresh snow bonds poorly with crust and due to its higher density it is also more likely to slip from crust. When ruleis satisfied and little or no snowfall occurs, the triggering probability is higher than days when mean snow height is much higher. This suggests significant melting instability. Temperature trend within these bounds can be another indicator of instability: warming trend can decrease stability, cooling trend can increase stability. In March, temperature usually shows a warming trend (refer Figure 2).

The model inferred the effect of a critical snowpack feature (melt-freeze crust) from meteorological data. Capturing more complex snowpack properties e.g. persistence and strength of buried weak layers requires further feature exploration. Effect of persistent snowpack structures and climatic oscillations on avalanche activity has been analysed in detail by many researchers (Laternser and Schneebeli, 2003; Hägeli and McClung, 2003; Thumlert et al., 2014). The resulting




**Table 7: Summary of comparisons with other model (HSS scores strongly depend on the training and testing datasets used).**

| Modeling technique used | Highest score achieved | Training data used | Automatic measurement of all features | Descriptive output |
|---|---|---|---|---|
| Support Vector Machines (Pozdnoukhov et al., 2008, 2011) | HSS = 0.62 TSS = 0.63 | 10 years data (1991 – 2000), Lochaber region (Scotland) | No | Explored support vectors of hyperplane of trained SVM |
| Calibrated nearest neighbours (Singh et al., 2014) | HSS = 0.31 | 14 years data (1999 – 2012), Chowkibal-Tangdhar axis (India) | No | Returns a list of similar days and their attributes measured by calibrated metric |
| Calibrated nearest neighbours (Purves et al., 2003) | TSS = 0.61 | 8 years data (1991 – 1998), Lochaber region (Scotland) | No | Returns a list of similar days and their attributes along with graphical visualisations of attributes and geo-map locations of similar days |
| Random Forest (This model). | HSS =0.42 TSS = 0.47 | 3 Years data (2010 – 2013), BG sector (India) | Yes | Displays decision trees in ensemble predicting high avalanche probability |

characterisations of avalanche climates can be used to derive relevant indexes to forecast (Haegeli and McClung, 2007; Shandro and Haegeli, 2018). This model can be expected to account for these complex effects using simple and relevant

extracted features.

Following are examples of some other frequently used avalanche forecast models with descriptive outputs:

(a) Nearest neighbours model lists similar days and their attributes (Buser, 1983; Purves et al., 2003; Singh et al., 2014).

(b) Expert systems list applicable rules (Schweizer and Föhn, 1996).

(c) Support Vector Machines can list vectors which define the maximal margin hyperplane (Pozdnoukhov et al., 2011).

The descriptive output of *k*-nearest neighbours model is a list of most similar days to the day being forecasted. From this list forecaster makes inferences about important variables contributing and unstable slopes. However, understanding interactions between variables is difficult using this approach since numerical data about variable combinations causing

hazard is unavailable. Forecasters have to use only few similar days, therefore variable interactions are deduced from experience largely. In contrast, visuals of trees can show important interactions between variables and give useful numerical data. Trees show the critical variables for a day and the range of values of these variables which were historically related to avalanche hazard. The path of a decision tree can be interpreted as a forecasting heuristic with confidence estimates from past data.


**6. Comparisons with other models**

The model has reasonably good classification performance (refer Table 5) given the difficulty of forecasting natural avalanches. Even the false alarms may indicate un-triggered snow instability. Descriptive output can provide more information about nature of these instabilities and their probable locations. The model uses lesser data and has potential for

complete automation of decision making process for avalanche forecasting .Sufficient historical data is not available for many places therefore a data efficient model may prove to be quite helpful for avalanche forecasting in such places. We compared the proposed model with other available avalanche forecasting models based on skill scores, selection of



features, data efficiency, potential for automation and descriptive output features. A summary of comparison is presented in Table 7.


### 7. Data Efficiency and Potential for Autonomous Process

RF model demonstrated promising performance over four consecutive winter seasons while the data of only three winter seasons was used for model construction (refer Section 4). An explanation of this data efficiency is that while decision trees model decision reasoning, the ensemble accounts for the different causes of avalanches. Different features cause

avalanche hazard under different situations. Therefore, the features involved in causing hazard vary across the sample space. Trees in ensemble can account for the important contributing features under different situations. The trees trained on features matching the contributing features for input day have higher probability outputs than other trees. Nearest neighbour models are unable to adapt to this variation in feature importance since they use the same distance metric to forecast in every neighbourhood of sample space.

The proposed ensemble prediction is based on features which can be measured automatically. Therefore such models can use data from dense sensor grid to improve performance. If new features are included to improve modelling process, only a few records of these new features are required for training an updated model. Therefore data efficiency of a model also implies economic sense from the point of view of setting up or updating a sensor grid.

The model's ability to discern different situations (even the likes of situations not encountered before) with reasonable

accuracy due to its sophisticated learning process, data efficiency and amenity to automatic observations allow for setting-up autonomous avalanche forecasting frameworks with minimum human intervention. Such set-ups can be useful for management of transport corridors sections and ski-slopes etc.

### 8. Conclusions and future work

Requirement of long term training data is a significant problem in operational use of machine learning models for avalanche forecasting. Data efficiency can reduce the cost of training a new model for a location or retraining an existing model to use different data. This paper demonstrates the use of Random Forest technique for avalanche forecasting on a dataset from an avalanche prone area in north-western Himalaya in India. The model demonstrated reasonable forecast skill while using low amount of training data. This is likely due to the ability of decision trees to model specific avalanche

forecast knowledge and of ensemble to model the stochastic properties of data. Data used by model for prediction can be collected automatically. Automated data collected in high volume from a dense sensor grid can be used for generating localized forecasts. With these properties, the model has potential for implementing an autonomous decision making framework. Future research can explore reducing the data requirements further by using transfer learning techniques (Pang and Yang, 2010).

Descriptive outputs explain reasoning for predicted avalanche hazard level and help forecasters' judgement by giving them probability estimates and qualitative analysis of situation. The reasoning given by the model can help forecasters validate their assumptions about the current situation or alert them if these assumptions are invalid. Features combinations causing hazard and avalanche probabilities given the feature ranges are easily comprehensible from decision trees. Further data mining can be attempted using these ranges and features to find unstable slopes and type of instabilities. For other models

presently being used, such conditions need to be inferred manually causing more subjective bias.

### Code Availability

None

### Data Availability



None

**Sample Availability**

None

**Video Supplement**

None

**Appendices**

None

**Author contributions**

*ManeshChawla:* Implementation and validation of model; Data exploration and visualisation; Writing manuscript.

*Amreek Singh:* Help with writing Section 2, 3 and 4; Suggestions for model improvement; Model performance analysis;

Review of manuscript.

**Competing Interests**

Authors are employed by Defence Geo-informatics Research Establishment, Defence Research and Development organisation, Ministry of Defence, Government of India.

**Disclaimer**

The information contained in this paper is true and complete to the best of our knowledge. The authors disclaim any liability in connection with the use of this information.

**Acknowledgements**

Authors are grateful to Internal Paper Review Committee and Director, Snow & Avalanche Study Establishment(now

Defence Geo-informatics Research Establishment) for review and permission to publish this paper. We are thankful to various field workers of SASE for data collection and maintaining the field equipment in difficult weather conditions. HS Sodhi contributed in preparation of Figure 2. We thank SASE data management team for providing necessary data used in the study. Graphs were generated using *Microsoft Excel* and *Python matplotlib*.

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
