# Peer review of "A data efficient machine learning model for autonomous operational avalanche forecasting"

_Natural Hazards and Earth System Sciences, 2021_

## Referee Comment (RC1)

**Review of "A data efficient machine learning model for autonomous operational avalanche forecasting" by Chawla and Singh**

General Comments

The authors present a study that uses an avalanche occurrence dataset from The Great Himalaya Range in northwest India during the period 2010-2013 to train a Random Forest model. They then test this model on avalanche occurrence data from 2013/14 to 2017. Overall, the work is interesting and highlights the usefulness of a statistical model in avalanche forecasting from data sparse regions. The authors appear to apply the techniques appropriately and the use of Random Forest modelling, while not new in avalanche research, provides further evidence that it could potentially be incorporated as another tool in avalanche forecasting operations. However, there are a few issues that need to be addressed prior to publication.

First, the manuscript requires further restructuring. The Introduction is thorough and clear up until Section 4 where methods are mixed with model tuning results, and it is difficult to keep track of methods and results. I suggest having distinct Results sections on model performance and model generalization that are separate from the description of the implementation of the model (i.e. methods). This should make the results clearer to the reader.

Second, the authors state that the advantage of the ensemble RF model is due to the ability of the model to provide information on the stochastic nature of the process. The authors provide evidence of individual trees detailing the line of reasoning, but I don't see interpretation of the physical processes described by the overall ensemble. The authors detail the predictive capabilities of the model by presenting POD, FAR, and HSS scores and provide examples of specific trees within the ensemble (which is good), but not how the overall ensemble provides insight into the physical processes at play. Presenting the overall variable importance and describing what this means for avalanche occurrence, in general, in this region would be helpful. The two specific examples provided in Figures 8 and 9 are a nice way to visualize individual days, but a clearer explanation of how the overall ensemble model describes the general processes at play and enhances avalanche forecasting is necessary.

Finally, there are several items detailed below in the Specific Comments section that should be addressed to help clarify certain issues. Most importantly, I think more detail on the avalanche dataset is necessary (see comment below).

Specific Comments

line 82: It would be very useful to incorporate how RF models have already been used in the avalanche literature in this section (e.g.(Mitterer and Schweizer, 2013;Guy and Birkeland, 2013;Marienthal et al., 2015;Dreier et al., 2016)). Currently, there are no references to how RF has been used in avalanche research, specifically. In addition, results from your study compared to these studies should be interpreted in the Discussion.

lines 98-111: Classification trees and even RF (see comment above) have been used enough in avalanche research that this detailed description probably isn't necessary. Consider providing a succinct sentence overview of the process and directing the reader to Breiman, 2001 and other avalanche studies that use these techniques.

line 115-137: I appreciate the technical detail and description of the C4.5 algorithm RF method as I am not familiar with this specific RF approach. However, I suggest condensing all of Section 2 to a broad and succinct overview with appropriate references for the reader and then referring the reader to a Methods

supplement which contains the specific equations and more detail. Lines 138-144 already provide a start to the succinct summary of RF.

line 163: The avalanche occurrence dataset is very important in this study. Can you please describe in more detail how avalanches are observed? I assume that all avalanche observations are derived using infrasonic and seismic sensors since line 165 states "all necessary input data for the proposed model may be recorded using automatic devices". Is there some sort of manual observer network or at least some manual validation of the infrasonic and seismic signal? Also, how much of the study area is instrumented? In other words, what proportion of the area are you able to detect avalanches on a daily time step? The quality of the avalanche dataset will likely have a great influence on the results. So, some discussion here of the quality control and/or limitations of the avalanche dataset would be useful, particularly the signal validation component (i.e. false alarms from the infrasonic and seismic sensors).

line 170-172: Are these seasons similar in input variables? If so, was this tested statistically? I suspect that if the variables in the seasons in the training dataset are significantly different than the test dataset, then this would adversely affect the model results? Can you provide some insight on this?
Also, please provide sample size (n=?) for the training and test datasets, respectively.

line 273-275: This statement is confusing to me as currently written. Did you use the temperature values from lines 269-270 as thresholds? What are the "temperature bounds" in Table 6? It is unclear. Also, this should be presented in the methods, not the results. See General Comments re: restructuring.

line 277: Again, what exactly is the "temperature bound rule"?

Table 6: Please list the sample size of the full dataset (avalanche and non-avalanche days). Also, I assume that this is the dataset from the RF model output using N=5000, and not the original observed avalanche dataset. Please clarify this in the caption and text.

line 284: Please explain how the model is able to infer the formation of a melt-freeze crust? As I understand it the model provides the probability of avalanche occurrence based on the input parameters? Can you really extend this to mean that a M/F crust formed since this is not a snowpack model? Assuming the "temperature bound rule" uses the values presented earlier, it's not clear to me how we can infer the formation of a melt-freeze crust from these data/trees? Please provide a clearer line of reasoning/evidence to support this claim.

lines 291-304: This section seems out of place and should probably be placed in the Introduction.

line 313: Can you discuss how your results actually compare to other studies and provide some interpretation on this rather than presenting the table?

line 329: How can the model account for "situations not encountered before" if they don't exist in the training dataset?

Technical Corrections
Some of the sentence structure/wording is difficult to follow at times. Consider a grammar/language revision.

lines 104 and 107: change "till" to "until"

line 126-129: font size seems different

line 218: HSS is spelled out in Table 4, but should be done so in the main text body as well.

Figure 5: Please include "results from test dataset" in caption.

References

Dreier, L., Harvey, S., van Herwijnen, A., and Mitterer, C.: Relating meteorological parameters to glide-snow avalanche activity, Cold Regions Science and Technology, 128, 57-68, 10.1016/j.coldregions.2016.05.003, 2016.

Guy, Z. M., and Birkeland, K. W.: Relating complex terrain to potential avalanche trigger locations, Cold Regions Science and Technology, 86, 1-13, 10.1016/j.coldregions.2012.10.008, 2013.

Marienthal, A., Hendrikx, J., Birkeland, K., and Irvine, K. M.: Meteorological variables to aid forecasting deep slab avalanches on persistent weak layers, Cold Regions Science and Technology, 120, 227-236, 10.1016/j.coldregions.2015.08.007, 2015.

Mitterer, C., and Schweizer, J.: Analysis of the snow-atmosphere energy balance during wet-snow instabilities and implications for avalanche prediction, The Cryosphere, 7, 205-216, 10.5194/tc-7-205-2013, 2013.

---

## Referee Comment (RC2)

**A data efficient machine learning model for autonomous operational avalanche forecasting**

Michaela Wenner, Reviewer comments

May 2021

**1 General comments**

Chawla et al. present results from an automated avalanche risk assessment algorithm using machine learning (ML). The authors propose random forest (RF) as their ML-Algorithm of choice. RF is an ensemble learning method meanwhile widely used in the field of natural hazards. The authors thoroughly explain the method and benefits thereof. As input features they use intrinsic snow-meteorological features such as temperature, snow height, wind speed and occurring avalanches of the last 24 hours. Additionally, they derive further features from this data, such as the change in snow surface temperature and new snowfall as well as wind speeds of the last couple of days. They process in total seven years of data; three years to train the model and four years to test the model. They define two different classes: no-avalanche day and avalanche day. In order to counteract imbalanced data (more days without avalanches) they applied a filter to only consider days with a snow height larger than 0.5m. After tuning some hyper parameters (tree depth and number of trees) they show the Heidke Skill Score (HSS), False Alarm Rate (FAR), Probability of Detection (POD) and Precision results of the training process (5-fold-cross validation). They end up with a HSS score of max 0.33. When applying the model on the test data set (called generalization in the paper), the performance increases to a max HSS of 0.42. From thereon specific days and predictions are discussed and an example of a decision tree from this day is shown graphically. They conclude that random forest is superior to other ML techniques for avalanche hazard assessment due to its data efficient behavior and comprehensive feature analysis. It can therefore help forecasters to make a decision not only by the probability output of the model but also its "graphic" decision making process.

The study is well within the scope of NHESS and is interesting for the avalanche hazard community. The difficulty of predicting avalanches is a well known problem, therefore the low prediction accuracy does not come as a surprise.

A strong point of the study is surely the discussion on what the visualisation of a single RF tree can tell us about the prediction making process and the decision rules that were applied for each feature. As RF is a relatively simple ML algorithm this might encourage more people in the community to make use of automatic techniques to find decision making rules in a data set. Additionally the paper profits from a clear introduction.

However the paper suffers from several points: (1) Performance analysis: the authors refrain from setting a classification threshold which therefore does not allow to give clear evaluation of a confusion matrix and how well the model actually performs. There are ways to evaluate the performance more clearly, which I will mention further in the specific comments. Additionally, a more thorough feature analysis (over all trees) should be performed to find most important features on a RF level and not just a single decision tree level (2) A direct comparison to other ML algorithms is missing. The authors do compare the algorithm performance to other studies, however, the amount of data they use is very different. A simple performance comparison to another ML technique (such as k-nearest-neighbours) would give a much better insight on how capable RF is. (3) There are

several typos and unclear sentences with sometimes incorrect grammar and often incorrect punctuation/white spaces. Therefore, before publication can be granted, I recommend a thoughtful review of the aspects detailed below.

**2 Specific comments**

The paper suffers from an unclear structure, especially in the methodology, results and discussion part. I recommend restructuring the paper according to the conventional structure: Start out with explaining every method used in this paper, then show and describe the results obtained and in a last step discuss what this might mean. Specifically for this paper I would recommend moving section 3 to directly after the introduction, then continuing with a methods part which can be subdivided into the description of random forest, then the data pre-processing, performance measures, model training and hyper parameters tuning and model generalization. Then show and describe the results of above mentioned methods. This means to basically describe Figures 4-7. In the discussion part the authors can then mention the model output interpretability and data efficiency and potential for autonomous process.

Another aspect which would greatly improve the paper is a more extensive performance analysis. I suggest using receiver operating characteristics (ROC) and the area under the curve (AUC) to evaluate the model performance additionally. This way, one can directly see the trade-off between the true positive rate and false positive rates. Then I suggest to define a threshold and include a confusion matrix in the performance analysis. I understand that in the end the threshold value is for the "operator" to define, but I think for this study it would benefit to give an example threshold and evaluate the model with a confusion matrix based on this value. I do like the probability presentation in Figure 5, but I think an additionally confusion matrix is needed. Furthermore, instead of only showing single trees, a feature importance analysis over the whole forest would greatly improve the manuscript.

In line 291-304 and table 7 the authors compare their model to other studies using different ML techniques. However, the amount of training data used in other studies is quite different to what has been used in this case. This of course makes it hard to actually compare the performance between two ML algorithms. I think it would be beneficial for this study to compare the results obtained with RF to another ML technique, e.g., support vector machines or k-nearest neighbours. This way, the authors can describe (1) why they prefer RF as techniques - of course also because of its interpretability and (2) how the amount of training data influences the performance - in comparison with the other studies and (3) explain differences to other studies, e.g., in the features that were used.

Line 326-329 the authors state that new features can be added easily to improve the model. I do not understand completely how this is done. Please explain more excessively and give references.

Please remake Figure 2. Instead of only showing the medians I suggest to use boxplots for a clearer picture of climatic conditions at the study area.

Add a few sentences in the discussion on how the model could be generalized for example for other sites. Is it possible to use one and the same model for all sites? Or sites at similar altitudes? Please discuss.

**3 Technical corrections**

**3.1 Abstract**

l11 Delete "world over"

**3.2 Introduction**

l28 "e.g.," comes with a comma behind the second dot. This is forgotten several times throughout the manuscript

l33 Please add white spaces. Also this is encountered quite often

l45 Please rephrase the sentence.

l59-65 Please clarify what is meant here.

l78 If you mention detection of avalanches using seismic data, you might add our recent paper (Wenner et. al 2021) in which we actually also use RF to detect mass movements (also avalanches)

l83 Maybe rephrase to "(...) for binary classification (avalanche day – no-avalanche day)

l87-88 Is the input not the same as for other ML models? Please clarify what you mean with this statement

l90-95 As mentioned in the specific comments, I think a restructuring of the paper would benefit the overall reading experience

**3.3 Random Forest Technique**

l104 Please write "until" instead of "till" (same for line 107)

l108-111 Please rephrase and clarify

**3.4 Study Area and Data Characteristics**

l148 In the north-western part of "the" Indian Himalaya

Table 1 "Intrinsic snow-meteorological features used by "the" model" (I suggest to include a "the" before model)

**3.5 Data pre-processing**

l171 Leave out "only"

Table 3 I would find it more informative to see the total number of avalanches days instead of the mean, and accordingly total number of avalanches that happened with a snow height below 0.5m

Table 4 I like the table, it gives a great overview. Maybe you could just add in the caption what i,j are (true label vs classifier label)

**3.6 Performance measures**

l186 Here, the confusion matrix is described, but then not used in the paper. I strongly suggest to do that though (as mentioned in specific comments)

**3.7 Model training and hyper-parameter tuning**

Figure 4 Add standard deviation from 5-fold-cross-validation. Additionally, make sure that the term "5-fold-cross-validation" is equally written in the text and the figure title

l197 Explain quickly what cross-validation is and what the 5-fold means

l198 There are more than two hyper-parameters for RF. Explain why you didn't change those or rather why you set it to this value.

l202 Please rephrase and clarify this statement

l205-209 This should definitely go to the discussion section of the paper

l210-212 Delete "Based on this premise" and rephrase sentence
l226 Yes, but this is intrinsic with the definition of both. How about the the POD? Also, please address Figure 4a and Figure 4b separately and explain what is shown.

l232 What would those complex situations be? Please discuss (in the discussion section)

l233 Consider using normal words instead of the feature names - new snow instead of new_snow (the connection is easy enough)

**3.8 Model generalisation**

l239-241 Consider rephrasing the sentence to: "After hyper-parameter tuning through cross-validation experiments for optimized performance, we constructed (...)"

**3.9 Model Output Interpretability**

l254 How can you find unstable slopes with that? Please clarify

l262 Consider rewriting to "(...) high wind speed as the most indicative feature for avalanche hazard"

l280-281 Please clarify the sentence

l283 I suggest to skip the last sentence

l290-304 I like the comparison to other studies, however I think it could be a bit more comprehensive. Maybe rephrase and make sure to clarify your message. Also, this should be in the discussion section.

**3.10 Comparison with other models**

l309 "The model uses lesser": what do you mean by lesser? In terms of quality or quantity?

**3.11 Data Efficiency and Potential for Autonomous Process**

l326-328 Please rephrase and clarify

**4 References**

Wenner, M., Hibert, C., van Herwijnen, A., Meier, L., and Walter, F.: Near-real-time automated classification of seismic signals of slope failures with continuous random forests, Nat. Hazards Earth Syst. Sci., 21, 339–361, https://doi.org/10.5194/nhess-21-339-2021, 2021.

---

## Author Comment (AC1)

**Reply to Erich Peitzsch**
**June-, 2021**

Thank You for presenting a detailed review of the manuscript. Your comments will improve the manuscript greatly.

**1. Reply of General Comments**

**1.1 Restructuring**

The manuscript requires further restructuring. The Introduction is thorough and clear up until Section
4 where methods are mixed with model tuning results, and it is difficult to keep track of methods and
results. I suggest having distinct Results sections on model performance and model generalization that are
separate from the description of the implementation of the model (i.e. methods). This should make the
results clearer to the reader.

Presently section 4 of manuscript includes description of methods used for performance measures, cross-validation and model generalisation. As suggested, these can be included in a separate methods section. Accordingly, following restructuring changes can be made in the revised manuscript:

1. Section 4.2 [Performance measures] can be moved into methods section.

2. From Section 4.3 [Model training and hyper parameters training] the details of hyper parameter training can be moved into a sub-section under methods section.

3. From Section 4.4 [Model Generalisation ] the details of how the model was trained and  tested will be moved to a sub-section of methods section. These details will only be referred from the section on model generalisation.

**1.2 Interpretation of the physical process described by overall ensemble**

The authors state that the advantage of the ensemble RF model is due to the ability of the model to
provide information on the stochastic nature of the process. The authors provide evidence of individual
trees detailing the line of reasoning, but I don't see interpretation of the physical processes described by
the overall ensemble. The authors detail the predictive capabilities of the model by presenting POD, FAR,
and HSS scores and provide examples of specific trees within the ensemble (which is good), but not how
the overall ensemble provides insight into the physical processes at play. Presenting the overall variable
importance and describing what this means for avalanche occurrence, in general, in this region would be
helpful. The two specific examples provided in Figures 8 and 9 are a nice way to visualize individual
days, but a clearer explanation of how the overall ensemble model describes the general processes at play
and enhances avalanche forecasting is necessary.

We have computed the importance score for each training feature, presented in the bar graph below.

[Figure]

Some observations:

1. SNOW_HEIGHT has the highest contribution in avalanche formation followed by the variable for cumulative snow-fall of past few days.

2. NEW_SNOW has low contribution, yet the cumulative snow fall history is important(NEW_SNOW10, NEW_SNOW4).

3. AVAL has low contribution, yet the cumulative avalanche history is important ( AVAL4 , AVAL2).

From 2 and 3, we may believe that variables showing past 4 day snow instability ( AVAL4 ) and 4 day snow fall are more valuable for avalanche forecasting than the variables showing immediate instability and snowfall. The weather and snowpack history of past few days contributes in complicated ways to increase hazard. We will analyse this in future work.

**2. Specific Comments**

line 82: It would be very useful to incorporate how RF models have already been used in the avalanche literature in this section (e.g.(Mitterer and Schweizer, 2013; Guy and Birkeland, 2013;Marienthal et al., 2015; Dreier et al., 2016)). Currently, there are no references to how RF has been used in avalanche research, specifically. In addition, results from your study compared to these studies should be interpreted in the Discussion.

These references will be included and discussed in our revised manuscript.

lines 98-111: Classification trees and even RF (see comment above) have been used enough in avalanche research that this detailed description probably isn't necessary. Consider providing a succinct sentence overview of the process and directing the reader to Breiman, 2001 and other avalanche studies that use these techniques.

This can be done in the revised manuscript, however referee 2 asked us to rewrite this clearly. Hence we propose to retain detailed description (though in a revised manner as suggested by refree 2)

line 115-137: I appreciate the technical detail and description of the C4.5 algorithm RF method as I am not familiar with this specific RF approach. However, I suggest condensing all of Section 2 to a broad and succinct overview with appropriate references for the reader and then referring the reader to a Methods supplement which contains the specific equations and more detail. Lines 138-144 already provide a start to the succinct summary of RF.

As suggested, the Section 2 can be further condensed and shifted under 'Methods' section.

line 163: The avalanche occurrence dataset is very important in this study. Can you please describe in more detail how avalanches are observed? I assume that all avalanche observations are derived using infrasonic and seismic sensors since line 165 states "all necessary input data for the proposed model may be recorded using automatic devices". Is there some sort of manual observer network or at least some manual validation of the infrasonic and seismic signal? Also, how much of the study area is instrumented? In other words, what proportion of the area are you able to detect avalanches on a daily time step? The quality of the avalanche dataset will likely have a great influence on the results. So, some discussion here of the quality control and/or limitations of the avalanche dataset would be useful, particularly the signal validation component (i.e. false alarms from the infrasonic and seismic sensors).

All the snow-meteorological and avalanche observations used in this work were observed and recorded manually using traditional instruments and visual observations. The details of snow-met observatory is given in line 154. The observatory is manned by trained persons. The avalanche occurrences in the remote sections of study area are reported by the villagers and Indian Army troops manning the border outposts. It is likely that some occurrences might have passed unreported. As rightly pointed out that this has an implication on the quality of dataset and in turn on the results. But this limitation about avalanche occurrence observations is well recognised by avalanche research community. However, the point that we wanted to convey by line 162-166 is that all these features can also be recorded automatically as the technology for the same exists. If this technology is adopted, it may remove dependency on manual presence to observe and record data (snow-met as well as avalanche occurrences). In other words, the data requirements of proposed model can be met with commercially available automatic systems. At this juncture, the issue of data quality is beyond the scope of the subject matter of this paper, as the paper focuses on the RF as a technique for decision making, interpretability of results in terms of physical process responsible for the event, and the potential it carries to be used as an autonomous tool. However, in order to avoid any confusion and make the point more clear, the paragraph (line 162-166) will be re-written in revised manuscript.

line 170-172: Are these seasons similar in input variables? If so, was this tested statistically? I suspect that if the variables in the seasons in the training dataset are significantly different than the test dataset, then this would adversely affect the model results? Can you provide some insight on this? Also, please provide sample size (n=?) for the training and test datasets, respectively.

These seasons are not similar in all input variables. The testing dataset (Dec 2013- Mar 2017) (n=485) carries higher temperature and lower standing snow than the training dataset (Dec 2010-Mar 2013) (n=364). This was tested statistically. We believe this does not affect the model performance. Our model gives the conditional probability of an avalanche given the input variables, the relation of avalanches on the factors should stay the same even if the distribution of those factors changes during the seasons. For example:

Number of days in training dataset with snow-fall > 20cm was: 36
Number of days in testing dataset with snow-fall > 20cm was: 26

We observe that snow-fall days were fewer in the testing dataset, despite it being of longer duration than training dataset. Yet the probability values of an avalanche occurring when snow-fall > 20 cm were found to be 0.55 for training dataset and 0.5 for testing dataset which are comparable.

line 273-275: This statement is confusing to me as currently written. Did you use the temperature values from lines 269-270 as thresholds? What are the "temperature bounds" in Table 6? It is unclear. Also, this should be presented in the methods, not the results. See General Comments re: restructuring

We filtered the dataset using the following rule:

$-2.75^{\circ}C < MIN\_TEMP \leq -0.75^{\circ}C \ \ AND \ \ MAX\_TEMP \geq 1.75^{\circ}C$

The bounds on MIN_TEMP and MAX_TEMP have been called temperature bounds and have been adopted from Figure 9. In this filtered dataset we found a greater percentage of avalanche days than the original dataset. The increase in avalanche hazard may have been caused by other factors which occur more frequently within these temperature bounds. We did a univariate analysis to rule out such factors: if the distribution of a variable in the filtered and unfiltered datasets differed, it might be contributing in increasing the hazard when the temperature bounds are satisfied. Therefore such variables were further analysed and results compiled in Table 6.
The above analysis is justified in 'results' section and not 'methods'.

line 277: Again, what exactly is the "temperature bound rule"?

A day satisfies the temperature bound rule if the MAX_TEMP and MIN_TEMP recorded that day satisfy the following:

$-2.75^{\circ}C < MIN\_TEMP \leq -0.75^{\circ}C \ \ AND \ \ MAX\_TEMP \geq 1.75^{\circ}C$

Table 6: Please list the sample size of the full dataset (avalanche and non-avalanche days). Also, I assume that this is the dataset from the RF model output using N=5000, and not the original observed avalanche dataset. Please clarify this in the caption and text.

This is the complete dataset (training + testing) from Dec 2010 – Mar 2017. The size (n) of samples used for each estimate has been mentioned in the table.

Including the training dataset from which we derive the rule may bias the results to support the rule. Therefore we have made an additional table using only the testing dataset as well. The table is consistent with all the conclusions drawn from Table 6. We have given this table in appendix section of this reply.

line 284: Please explain how the model is able to infer the formation of a melt-freeze crust? As I understand it the model provides the probability of avalanche occurrence based on the input parameters? Can you really extend this to mean that a M/F crust formed since this is not a snowpack model? Assuming the "temperature bound rule" uses the values presented earlier, it's not clear to me how we can infer the formation of a melt-freeze crust from these data/trees? Please provide a clearer line of reasoning/evidence to support this claim.

A day satisfies the temperature bound rule if the MAX_TEMP and MIN_TEMP recorded that day satisfy the following:

$$-2.75^oC < MIN\_TEMP \leq -0.75^oC \ \ AND \ \ MAX\_TEMP \geq 1.75^oC$$

If a day satisfies this rule then it has a sub-zero night temperature and above-zero temperature during the day. Melting is caused during day when temperature is greater than zero. The melted water on surface then refreezes again during night to form a crust.

Formation of M/F crust explains why the decision tree in Figure 9 shows high avalanche probability. Deduction of M/F crust from input features requires knowledge of physics not present in model, but in this situation the reasoning used by model is consistent with reasoning based on physical effects: M/F crust and higher snow density. The model was able to infer the effect of these physical effects without using information about them directly as we have not included the M/F crust or snow density in the input features set.

This will be corrected in revised manuscript.

Comparison using skill scores is provided in the table. We have also provided discussion on descriptive forecasts and data efficiency etc. vis-à-vis other models.

line 329: How can the model account for "situations not encountered before" if they don't exist in the training dataset?

By "situations not encountered before" we mean situations to which the model generalises from the training set e.g: there may be no situations during the training when snow-height > 3m, but we can expect the model to generalise to such situations. But model may not generalise to all future situations. This may be due to very few examples near similar to that situation, poor model bias, missing data etc.

**3. Technical Corrections**

lines 104 and 107: change "till" to "until".

line 126-129: font size seems different

line 218: HSS is spelled out in Table 4, but should be done so in the main text body as well.

Figure 5: Please include "results from test dataset" in caption.

These will be corrected in revision.

**4. Appendix**

| Statistic | Unfiltered Dataset | Dataset Filtered by Temperature Bounds | Dataset Filtered by SNOW_HEIGHT > 1.0m |
|---|---|---|---|
| **Proportion of Avalanche days** | 0.18 ( n = 485 ) | 0.33 ( n = 57 ) | 0.32 ( n = 164 ) |
| **Mean Snow Height** | 0.74m ( n = 485 ) | 1.13m ( n = 57 ) | 1.37m (n = 164 ) |
| **Proportion of Avalanche days when NEW_SNOW > 0 m** | 0.3 ( n = 169 ) | 0.51 ( n = 29 ) | 0.45 ( n = 78 ) |
| **Proportion of Avalanche days when NEW_SNOW > 0.1 m** | 0.35 ( n = 48 ) | 0.61 ( n = 18 ) | 0.43 ( n = 48 ) |
| **Proportion of Avalanche days when NEW_SNOW> 0.2 m** | 0.34 ( n = 55 ) | 0.58( n = 7) | 0.37 ( n = 27 ) |

**Table: A revision of Table 6 given in the manuscript, using only the testing data.**

---

## Author Comment (AC2)

Reply to Michaela Wenner
June, 2021

Thank You for presenting a detailed review of the manuscript. Your comments will improve the manuscript greatly.

**1. Reply of Specific Comments**

**1.1 Structure of manuscript**

In section 4 of manuscript we have included a description of performance measures, cross-validation and model generalisation which should be included in a methods section.
We believe that the data-preprocessing is very specific to the domain of avalanche forecasting and depends on our data-set, this sub-section should remain in Section 4 along with the results.

Following restructuring changes will be made in the revised manuscript:

1. Section 4.2 [performance measures] will be moved into methods section.
2. From Section 4.3 [ Model training and hyper parameters training] the details of hyper parameter training will be moved into a sub-section in methods section.
3. From Section 4.4 [ Model Generalisation ] the details of how the model was trained and the testing scores used will be moved to a sub-section of methods section. These details will only be referred from the section on model generalisation.

**1.2 Performance Analysis**

**1.2.1 Analysis using ROC curves and AUC scores**
Table 5 provides the exact FAR/POD trade-off found from the ROC curve i.e a sampling from the curve at uniform FAR intervals of 0.1. It provides additional scores to help readers compare the training and test performance. The AUC scores for testing and training phases are given in the appendix of this reply and will be provided in the caption of Table 5 (in the revision).

**1.2.2 Contigency Table Analysis**
The contigency tables can be reconstructed from the FAR and POD scores when the number of negetives (Total Negetives in formulas ) and positives (Total Positives) in the testing data is known.

$$POD = \frac{True\ Positives}{Total\ Positives} \qquad FAR = \frac{False\ Negetives}{Total\ Negetives}$$

Therefore using FAR,POD,  Number of Positives and Number of negetives the contigency table entries are:

$$True\ Positives = POD * Total\ Positives$$

$$False\ Positives = Total\ Positives - True\ Positives$$

$$False\ Negetives = FAR * Total\ Negetives$$

$$True\ Negetives = Total\ Negetives - False\ Negetives$$

We will provide the contigency tables for each FAR and POD level given in Table 5[in manuscript appendix]. These have been additionaly provided in the appendix of this document.

**1.2.3 Feature Importance Analysis**
We have computed the importance score for each training feature, presented in the bar graph below.

[Figure]

Some observations:

1. SNOW_HEIGHT has the highest contribution in avalanche formation followed by the variable for cumulative snow-fall of past few days.

2. NEW_SNOW has low contribution, yet the cumulative snow fall is important(NEW_SNOW10, NEW_SNOW4).

3. AVAL has low contribution, yet the avalanche history is important ( AVAL4, AVAL2 ).

From 2 and 3 , we may believe that variables showing past 4 day snow instability ( AVAL4 ) and 4 day snow fall are more valuable for avalanche forecasting than the variables showing immediate instability and snowfall. The weather and snowpack history of past few days contributes in complicated ways to increase hazard. We will analyse this in future work.

**1.2.4 Comparison with a simple baseline model to demonstrate data-efficiency**

We trained and tested a nearest neighbours model on the training and testing datasets of the RF model, the performance of RF model was found significantly better. The FAR/POD scores obtained and the AUC score of the NN-model are provided in appendix of this file.
These results will be discussed in comparisons section of the revised manuscript. They clearly show the data efficiency of RF model.

**2. Technical Corrections**

l78 If you mention detection of avalanches using seismic data, you might add our recent paper (Wenner et. al 2021) in which we actually also use RF to detect mass movements (also avalanches). This reference will be included in the revised version.

l87-88 Is the input not the same as for other ML models? Please clarify what you mean with this statement
Lines 87- 88 all the input parameters used for our model can be collected automatically. This is not true for many avalanche forecasting models, the comparisons section details this.

l108-111 Please rephrase and clarify
To learn a tree from the dataset an algorithm has to find the feature value and its threshold at each tree node. At each iteration the algorithm takes a dataset and gives a threshold (t), feature (f ) and two disjoint partitions of the dataset. One partition containes all data points where feature (f ) has values <= threshold (t), other contains all the data points where where feature (f ) has values > threshold (t). Algorithm starts with the entire datset initially, the split and feature found are recorded in the top-most node, the sub-datasets found are used to define the split and feature values of the right and left sub-childs. This leads to further splitting and a recursive definition of the tree structure.

Table 3 I would find it more informative to see the total number of avalanches days instead of the mean, and accordingly total number of avalanches that happened with a snow height below 0.5m
This information can be derived from information given in Table 3, in revised manuscript we will provide it for completeness.

Table 4: I like the table, it gives a great overview. Maybe you could just add in the caption what i,j are (true label vs classifier label)
This will be done in the revised manuscript.

186 Here, the confusion matrix is described, but then not used in the paper. I strongly suggest to do that though (as mentioned in specific comments).
The confusion matrices for all FAR levels given in Table 5 will be provided in the appendix of revised manuscript.

Figure 4 Add standard deviation from 5-fold-cross-validation. Additionally, make sure that the term "5-fold-cross-validation" is equally written in the text and the figure title

Please clarify the meaning of standard-deviation here. The results of the 5-fold-cross-validation depend only on the dataset and the classifier used, we are not choosing the 5 – folds randomly so the results dont change when we do it multiple times.

197 Explain quickly what cross-validation is and what the 5-fold means
This will be done in the methods section of revised manuscript.

198 There are more than two hyper-parameters for RF. Explain why you didn't change those or rather why you set it to this value.
Changing them did not result in significant performance difference in our experiments.

l202 Please rephrase and clarify this statement
We get the conditional probability of an avalanche occuring given the input parameters.

l205-209 This should definitely go to the discussion section of the paper
This will be done in the revised manuscript.

l226 Yes, but this is intrinsic with the definition of both. How about the the POD? Also, please address Figure 4a and Figure 4b separately and explain what is shown.
Line 225-226 The sentence "High classification threshold probability means only days when the model is highly confident are classified as positive (avalanche days)" will be rephrased to : "increasing classification threshold gives higher precision i.e the likelihood of an avalanche occurring on a predicted avalanche day increases.". The model gives fewer but more accurrate alarms when the threshold is increased, this leads to lower detection rates and higher precision scores.

l232 What would those complex situations be? Please discuss (in the discussion section)
Complicated situations involve factors which cannot be deduced with a high certainty from the input data alone e.g: buried weak layers, ice layers, depth hoar crystals. To account for the uncertainty involved we use the statistical modelling approach.

l233 Consider using normal words instead of the feature names - new snow instead of new snow (the connection is easy enough)
This will be done in revised manuscript.

l239-241 Consider rephrasing the sentence to: "After hyper-parameter tuning through cross-validation experiments for optimized performance, we constructed (...)"
This will be done in revised manuscript.

l254 How can you find unstable slopes with that? Please clarify
1. By analysing the samples in leaf node.

2. By identifying the important factors associated with the avalanche day, we can identify which slopes they will affect the most: e.g high temperatures affect south aspect slopes most, teperature gradients will affect slopes at a higher altitude most.

3. This can also be done by including terrain features when training the model.

l262 Consider rewriting to "(...) high wind speed as the most indicative feature for avalanche hazard"
This will be done in revised manuscript.

l280-281 Please clarify the sentence
Snow height is an important hazard factor, data analysis shows that avalanche probability is greater when snow height is higher. We found conditions ( temperature bound rule ) which causes the

hazard of lower snow height days to be greater than days with higher snow height which dont satisfy the conditions.

This will be done in revised manuscript.

We will move this into section 6 [ comparisons with other models ].

Quantity ( The model uses 3 year data, this will also be clarified by giving a comparison with a baseline nearest neighbour approach ).

Using more information about snow,weather and terrain parameters can improve avalanche forecast. This can be done by including additional features in models e.g: snow wetness index, snow stability index, satellite image features, terrain features etc. To use the new feature, a model must be trained from a dataset containing it. Data efficiency minimises the number of training records required that contain the new feature, this can help if the collection of feature was started recently e.g: by installing new sensor for previously unrecorded parameter.

**3. Appendix**

**Feature Importance From RF model.**

| | |
|---|---|
| WIND | 0.06 |
| NEW_SNOW | 0.03 |
| MAX_TEMP | 0.07 |
| MIN_TEMP | 0.08 |
| AVAL2 | 0.05 |
| NEW_SNOW10 | 0.08 |
| NEW_SNOW2 | 0.04 |
| NEW_SNOW4 | 0.08 |
| SNOW_TEMP | 0.05 |
| SNOW_TEMP_DIFF | 0.04 |
| SNOW_HEIGHT | 0.09 |
| WIND2 | 0.08 |
| WIND4 | 0.08 |
| AVAL4 | 0.05 |
| AVAL | 0.03 |
| WIND10 | 0.08 |

**NN-Classifier Performance: AUC Score NN[ 0.7 ], AUC Score RF: [ 0.82 ]**

| FAR | POD [NN ] | POD [RF] |
|-----|-----------|----------|
| 0.2 | 0.6 | 0.65 |
| 0.3 | 0.68 | 0.76 |
| 0.4 | 0.75 | 0.83 |
| 0.5 | 0.77 | 0.88 |
| 0.6 | 0.85 | 0.91 |
| 0.7 | 0.87 | 0.93 |

**Contigency Tables from RF model:**

| FAR [0.2 ] | Avalanche | No Avalanche |
|------------|-----------|--------------|
| Avalanche | 63 | 34 |
| No Avalanche | 78 | 310 |

| FAR [0.3 ] | Avalanche | No Avalanche |
|------------|-----------|--------------|
| Avalanche | 74 | 23 |
| No Avalanche | 116 | 272 |

| FAR [0.4] | Avalanche | No Avalanche |
|-----------|-----------|--------------|
| Avalanche | 81 | 16 |
| No Avalanche | 155 | 233 |

| FAR [0.5] | Avalanche | No Avalanche |
|-----------|-----------|--------------|
| Avalanche | 85 | 12 |
| No Avalanche | 194 | 194 |

| FAR [0.6] | Avalanche | No Avalanche |
|---|---|---|
| Avalanche | 88 | 9 |
| No Avalanche | 233 | 155 |

| FAR [0.7] | Avalanche | No Avalanche |
|---|---|---|
| Avalanche | 90 | 7 |
| No Avalanche | 272 | 116 |